# Two-Stage Link Loss Optimization of Divergent Gaussian Beams for Narrow Field-of-View Receivers in Line-of-Sight Indoor Downlink Optical Wireless Communication (Invited)

**Xinda Yan** [1], **Yuzhe Wang** [1], **Chao Li** [2,*], **Fan Li** [3,*], **Zizheng Cao** [2] and **Eduward Tangdiongga** [1]

1   Eindhoven Hendrik Casimir Institute, Eindhoven University of Technology, 5612 AZ Eindhoven, The Netherlands; x.yan@tue.nl (X.Y.); y.wang1@tue.nl (Y.W.); e.tangdiongga@tue.nl (E.T.)
2   Peng Cheng Laboratory, Shenzhen 518055, China; caozzh@pcl.ac.cn
3   School of Electronics and Information Technology, Sun Yat-Sen University, Guangzhou 510275, China
\*   Correspondence: lich03@pcl.ac.cn (C.L.); fan39@mail.sysu.edu.cn (F.L.)

**Abstract:** The predominant focus of research in high-speed optical wireless communication (OWC) lies in line-of-sight (LOS) links with narrow infrared beams. However, the implementation of precise tracking and steering necessitates delicate active devices, thereby presenting a formidable challenge in establishing a cost-effective wireless transmission. Other than using none-line-of-sight (NLOS) links with excessive link losses and multi-path distortions, the simplification of the tracking and steering process can be alternatively achieved through the utilization of divergent optical beams in LOS. This paper addresses the issue by relaxing the stringent link budget associated with divergent Gaussian-shaped optical beams and narrow field-of-view (FOV) receivers in LOS OWC through the independent optimization of geometrical path loss and fiber coupling loss. More importantly, the geometrical path loss is effectively mitigated by modifying the transverse intensity distribution of the optical beam using manipulations of multi-mode fibers (MMFs) in an all-fiber configuration. In addition, the sufficiently excited higher order modes (HOMs) of MMFs enable a homogenized distribution of received optical powers (ROPs) within the coverage area, which facilitates the mobility of end-users. Comparative analysis against back-to-back links without free-space transmission demonstrates the proposed scheme's ability to achieve low power penalties. With the minimized link losses, experimental results demonstrate a 10 Gbps error-free (BER < $10^{-13}$) LOS OWC downlink transmission at 2.5 m over an angular range of $10° \times 10°$ without using any optical pre-amplifications at a typical PIN receiver. The proposed scheme provides a simple and low-cost solution for high-speed and short-range indoor wireless applications.

**Keywords:** indoor OWC; divergent Gaussian beam; link loss; MMF





## 1. Introduction

Over the past five decades, the capacity of radio frequency (RF) wireless communications has experienced exponential growth, surpassing a million-fold increase. Currently, more than 85% of internet traffic is generated within indoor environments [1], resulting in an overwhelming demand that exceeds the capacity of the RF spectrum. As a prospective solution to alleviate this congestion, the utilization of higher-frequency regions within the electromagnetic spectrum for indoor wireless communication has garnered considerable anticipation [2].

The visible and infrared (IR) light spectrum offers a vast range of over 300 THz of unlicensed bandwidth, presenting an abundant resource. Due to the limited penetration capability of optical beams through opaque obstacles, the confinement of light within rooms or compartments provides an inexhaustible supply of bandwidth resources through wavelength reuse [3]. These distinctive advantages over RF-based counterparts have

sparked widespread interest in optical wireless communication (OWC) within the academic community. In the context of indoor OWC, whether operating in the visible or IR band, cost-effectiveness is a crucial consideration, emphasizing the need to avoid complex devices and high computational requirements. While the visible band combines communication and illumination with a constrained modulation bandwidth, the 1550 nm window in the IR band offers eye-safety up to 10 dBm optical power. Furthermore, it is compatible with existing fiber-optic networks and well-suited for high-speed wireless transmission. Presently, considerable research efforts have focused on single-device exclusive links in line-of-sight (LOS) OWC with narrow optical beams, achieving an impressive transmission rate of 112 Gbps per beam [4]. Two-dimensional beam-steering for narrow beams is commonly achieved through actively controlled elements, such as microelectromechanical systems (MEMS) [5] and liquid-crystal spatial light modulators (LC-SLMs) [6]. Nevertheless, precise tracking and steering techniques are essential to eliminate any misalignment between transceivers. A localization accuracy of 0.038° and a wide field of view (FOV) of 70° × 70° have been experimentally demonstrated by utilizing light detection and ranging (LiDAR) [7,8]. Nonetheless, the integration of LiDAR with OWC introduces complexity and high costs. To address the need for cost-effective and high-speed indoor wireless downlinks, Koonen et al. proposed a passive and compact two-dimensional arrayed waveguide grating router (AWGR) that enables equivalent infrared (IR) beam-steering without any moving parts [9]. By discretely tuning the wavelength, a slightly divergent optical beam scans a relatively large coverage area. However, achieving seamless coverage without interference between spatially adjacent users and considering the variance of received optical power (ROP) within the coverage area due to a Gaussian-shaped optical beam pose challenges. Alternatively, increasing the divergent angle of the optical beam can potentially cover the same area as AWGRs without any additional operations, albeit at the expense of increased link losses. Moreover, employing a ground glass-based diffuser to achieve this angular expansion naturally transforms the Gaussian-shaped beam generated by lasers into a flat-top beam [10]. Therefore, a more divergent beam holds the potential advantage of a homogenized intensity distribution compared to AWGRs. In this study, as elaborated by [11], sensitive and deterministic multi-mode fibers (MMFs) are manipulated within an all-fiber configuration to modify the transverse intensity distribution of the laser source, which plays a crucial role in optimizing either geometrical link loss or ROP homogenization. Notably, offset launch techniques flatten the averaged intensity distribution of speckle patterns when higher-order modes (HOMs) are predominantly excited [12,13].

This paper assumes an ideal Gaussian distribution for the divergent optical beam, with the transverse intensity distribution along the propagation axis characterized by geometric optics. The validity of these two assumptions is experimentally verified. Our analysis reveals two major types of optical power loss in divergent optical beams, namely geometrical path loss and fiber coupling loss, with an optical fiber coupled collimator serving as the light-gathering device for photodiodes. Simulation results indicate that geometrical path loss and fiber coupling loss are interdependent, resulting in an unacceptably high total link loss in short-range OWC applications. To address the stringent link budget in such scenarios of intensity modulation and direct detection (IM-DD), the independent optimization of geometrical path loss and fiber coupling loss is pursued. Specifically, the fiber coupling loss is mitigated through the use of a receiving collimator with adjustable focus. Essentially, this minimization can be achieved by altering the coupling distance between the receiving lenses and the optical fiber to compensate for focus shifts. As a result, the fiber coupling loss is eliminated, allowing for the separate optimization of these two significant loss factors. Conversely, the complete elimination of geometrical path loss in divergent beams is not feasible. Nonetheless, the transverse intensity distribution of the Gaussian beam can be modified using MMFs. By iteratively applying controlled perturbations to the MMF, adaptive shaping of the divergent optical beam can be achieved based on the specific locations of portable devices. Furthermore, the introduction of offset launch techniques leads to a homogeneous and optimized ROP. Consequently, the significant variation in

ROP experienced by portable devices within the coverage area is mitigated. It is worth noting that the proposed receiving structure with a narrow FOV effectively suppresses ambient light and multi-path distortion. In addition, the mobility of portable devices can be enhanced through the deployment of receivers with a larger FOV.

The remainder of this paper is organized as follows. Section 2 describes the basic formulas of Gaussian beams and parameters of optics used in simulations and experiments. In Section 3, the propagation and reception of the Gaussian beam is simulated. Section 4 evaluates the bit error rate (BER) performance of the proposed scheme in IM-DD using non-return-to-zero on-off keying (NRZ-OOK) modulation over the coverage area, and conclusions and discussions are presented in Sections 5 and 6, respectively.

## 2. Basic Formulas of Gaussian Beams and Parameters of Optics

By considering a slowly varying envelope (SVE) of the electric field, the Gaussian beam emerges as an analytical solution to the paraxial Helmholtz Equation [14]. In the context of indoor OWC, the transverse intensity distribution of single-mode fibers (SMFs) is presumed to adhere to an ideal Gaussian profile, with this Gaussian shape being preserved along its propagation axis in indoor environments [15]:

$$I(r,z) = I_0 exp\left(\frac{-2r^2}{\omega^2(z)}\right), \tag{1}$$

where $I_0$ is the peak intensity, $r$ is the transverse distance concerning the propagation axis $Z$, and $z$ is the axial distance. The expression indicates that the transverse intensity distribution of a Gaussian beam is circular and symmetric with no obvious boundaries and weakens with increasing transverse distance. More precisely, its intensity drops to $1/e^2$ of the peak intensity $I_0$ when $r = \omega(z)$, and $\omega(z)$ is typically referred to as the beam radius. The peak intensity $I_0$ can be expressed as:

$$I_0 = \frac{2P_o}{\pi\omega^2(z)}, \tag{2}$$

where $P_o$ is the total optical power emitted to the free space. Due to diffraction, the beam radius $\omega(z)$ keeps varying with $z$ as:

$$\omega(z) = \omega_0\sqrt{1 + \left(\frac{z}{z_R}\right)^2}, \tag{3}$$

where $\omega_0$ is the minimum beam radius along the propagation axis named beam waist, and it appears at axial distance $z = 0$. The wavelength and beam waist dependent Rayleigh length can be calculated by Equation (4):

$$z_R = \frac{\pi\omega_0^2}{\lambda}, \tag{4}$$

where $\lambda$ is the operating wavelength of the laser source, and the Rayleigh length represents the ability of optical beams to maintain collimation along their propagation direction. Equation (4) indicates that a larger divergent angle can be obtained by shrinking $\omega_0$, and vice versa. Herein, the target divergent angle of the optical beam is obtained by focusing a collimated beam.

In order to analyze the propagation characteristics of the divergent Gaussian beam within the context of short-range indoor OWC, a comprehensive set of simulation and experimental results is provided in the subsequent sections. The optical parameters employed in both the simulations and experiments remain consistent, and they are detailed in Table 1. Specifically, the emitting collimator holds a fixed focal length, while the receiving collimator features an adjustable focal length.

**Table 1.** Parameters of the optics used in simulations and experiments.

| Optics | Size [1] | FL [2] | CA [3] | Beam Waist | NA [4] |
|---|---|---|---|---|---|
| Fixed Col. [5] | 24 mm | 37.20 mm | X | 3.5 mm | 0.24 |
| Lens [6] | 25 mm | 20.00 mm | 22.5 mm | X | 0.54 |
| Zoom Col. [7] | 1.2-inch | Adjustable | 20.5 mm | X | 0.25 |

[1] Outer Diameter; [2] Focal length; [3] Clear Aperture; [4] Numerical Aperture; [5] Collimator with a fixed focal length; [6] Focusing lens; [7] Collimator with a adjustable focal length.

## 3. Simulation Results of Gaussian Beams

After a laser-fed SMF passes through the emitting collimator, the beam radius evolution of a Gaussian beam over 200 m is shown in Figure 1. At this range, the Gaussian beam gradually approaches at $y = kx$, where $k = \omega_0 / z_R$. In this case, the Gaussian beam can be seen as a point source at $z = 0$ with a half-divergent angle $\theta = \omega_0 / z_R$ in rad.

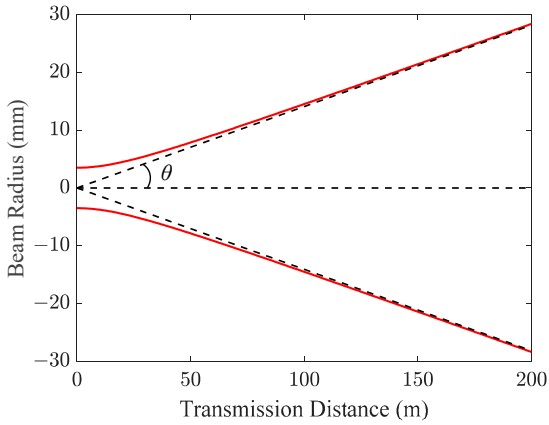

**Figure 1.** Beam radius evolution of the collimated beam within 200 m.

Figure 1 depicts the passage of the collimated beam through the divergent lens, and the corresponding $\theta$ is 0.14 mrad. In contrast to transmission distances spanning hundreds of meters, indoor OWC only requires a few meters. Figure 2 depicts the focusing process of the collimated optical beam after passing through the divergent lens. The emitting collimator's output facet is positioned at $z = 0$, while the thin lens marked by an arrow, neglecting its thickness, is situated at $z = 40$ mm. Since the separation between the emitting collimator and the divergent lens is constrained to several centimeters, the optical beam retains its collimation. As a matter of fact, its collimation distance is sufficient to extend a few meters away, and this characteristic accounts for the negligible geometrical path loss observed in indoor OWC employing narrow optical beams.

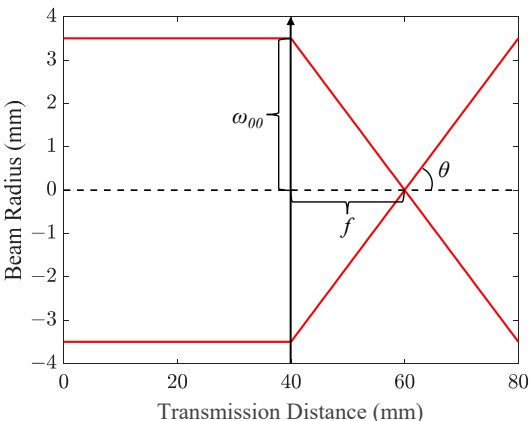

**Figure 2.** The focusing process of a collimated optical beam.

The divergence of the optical beam can be adjusted by lenses. As the optical beam encounters a lens, the beam waist of this lens is expressed as:

$$\omega_{01} = \frac{\omega_{00} f}{z_R},$$ (5)

where $\omega_{00}$ and $\omega_{01}$ are the beam waist of the emitting collimator and the divergent lens with a focal length of $f$, respectively. $z_R$ is the Rayleigh length of the emitting collimator. After the optical beam reaches its beam waist of 2.81 um as calculated by Equation (5), it diverges at the same rate as focusing, and the corresponding half-divergent angle $\theta$ is 9.92°, which equals the divergent angle calculated by $atan(\omega_{00}/f)$. As a result, the beam radius evolution, as determined by Equation (3), can be effectively substituted with a simplified geometric optics approach.

Following free-space transmission, the optical beam is collected by the receiving collimator. During this stage, geometrical path loss arises when the diameter of the coverage area exceeds that of the receiving collimator's clear aperture (CA). To evaluate the geometrical path loss, it is critical to establish the receiving model of divergent Gaussian beams. At a target transmission distance $z$, the optical power within transverse distance $r$ of a Gaussian beam is the integral of Equation (1) from 0 to $r$:

$$P(r,z) = \int_0^r I(r,z)dr = P_o\left(1 - exp\left(\frac{-2r^2}{\omega^2(z)}\right)\right),$$ (6)

where $\omega(z)$ is the radius of the coverage area at a transmission distance $z$. Assuming the receiving collimator with a CA of $\varphi_c$ is located at a transverse distance $r$, and the intensity distribution between the two circles with an on-axis dot and tangent to the receiving collimator is uniform [16], then the optical power captured by the receiving collimator can be derived from Equation (6) as:

$$P_{colli}(r,z,\varphi_c) = \begin{cases} P_o\left(1 - exp\left(\frac{-\varphi_c^2}{2\omega^2(z)}\right)\right), r = 0 \\ \dfrac{P_o\varphi_c\left(exp\left(\frac{-2\left(r - \frac{\varphi_c}{2}\right)^2}{\omega^2(z)}\right) - exp\left(\frac{-2\left(r + \frac{\varphi_c}{2}\right)^2}{\omega^2(z)}\right)\right)}{8r}, else \end{cases}$$ (7)

For a more intuitive look, the transverse distance $r$ is represented by receiving angle $atan(r/z)$ in the rest of the paper.

Figure 3a shows the geometrical path loss versus varying receiving angles within the coverage area at several short-range free-space transmission distances. Due to the Gaussian-shaped laser beam, the geometrical path loss scales up with the receiving angle. Regardless of transmission distances, the difference in geometrical path loss between the center (receiving angle = 0°) and the boundaries (receiving angle = ± 10°) of the coverage area is fixed at approximately 8.7 dB. In addition, the geometrical path loss is increased with the transmission distance at any receiving angle. Roughly, doubling the transmission distance will increase the geometrical path loss by 6 dB. Thereby, the geometrical path loss is one of the major link losses in divergent Gaussian beams.

The other significant link loss in the case of divergent optical beams is the fiber coupling loss. As depicted in Figure 4, an optical fiber is precisely positioned at the nominal focal length ($f$) of the receiving collimator. There is no focus shift ($\Delta f$) of the receiving collimator when its incident optical beam is collimated, and a divergent optical beam introduces a focus shift of the receiving collimator along the propagation direction.

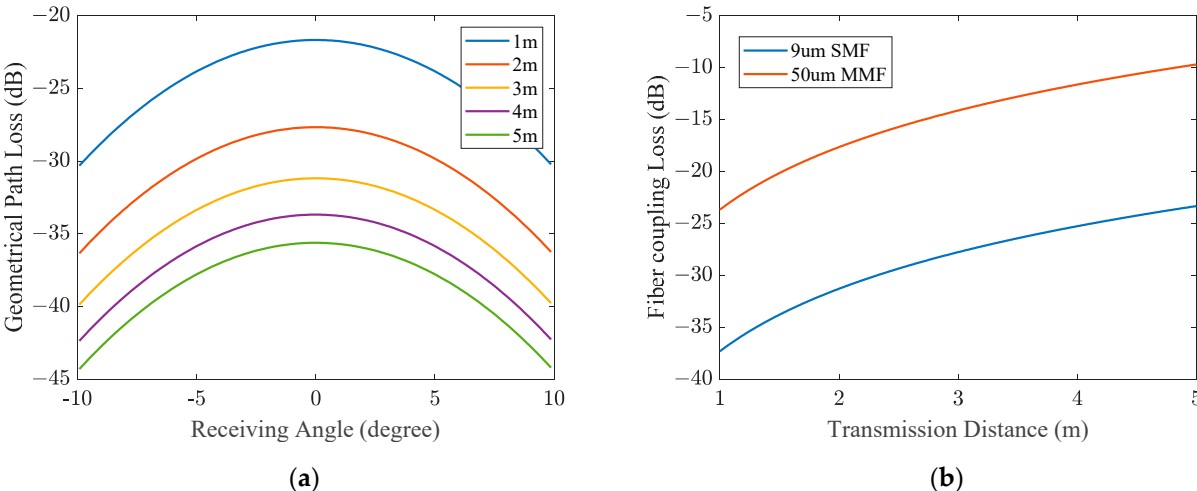

**Figure 3.** Two major types of optical link losses in divergent Gaussian beams. (**a**) Geometrical path loss versus receiving angles at different transmission distances; (**b**) Fiber coupling losses versus transmission distances using different fiber core diameters.

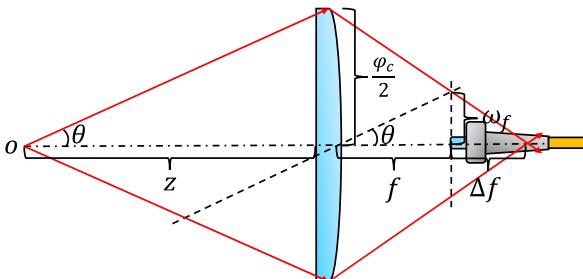

**Figure 4.** Fiber coupling of the receiving collimator.

According to geometric optics shown in Figure 4, the beam radius at the nominal focal length of the receiving collimator $\omega_f$ is:

$$\omega_f = \frac{f\varphi_c}{2z},$$ (8)

and the focus shift of the receiving collimator $\Delta f$ equals:

$$\Delta f = \frac{f\omega_f}{\varphi_c/2 - \omega_f},$$ (9)

substituting Equation (8) into Equation (9), we have:

$$\Delta f = \frac{f^2}{z - f},$$ (10)

in OWC applications, where $z >> f$, the first-order term of $f$ can be safely omitted. Figure 5a shows the focus shift of the receiving collimator when its focus length is set to the same as the emitting collimator (37.2 mm). When the transmission distance of the optical beam is 1 m to 5 m, the focus shift of the receiving collimator decreased from 1.43 mm to 0.28 mm monotonically. Figure 5b shows the beam radius at the nominal focal length of the receiving collimator. At the same transmission range, the beam radius decreased from 381 um to 76 um. Figure 3b illustrates the fiber coupling loss when using optical fibers with 2 typical core diameters without considering their numerical aperture (NA) limitation. It is evident that a greater transmission distance corresponds to a reduced fiber coupling loss, which stands in contrast to the geometrical path loss. Thus, optimizing

the link loss requires a careful consideration of the tradeoff between geometrical path loss and fiber coupling loss.

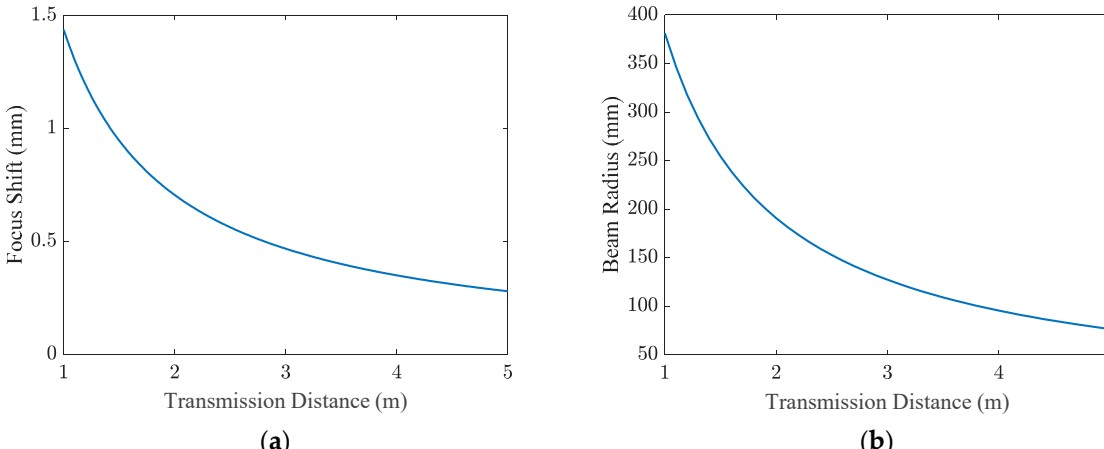

**Figure 5.** Focus shift of the receiving collimator and beam radius at the focus of the receiving collimator. (**a**) Focus shift versus transmission distances; (**b**) Beam radius at the focal length of the receiving collimator versus transmission distances.

The total link loss of the proposed system is the product of $\alpha$ and $\beta$, where $\alpha$ and $\beta$ presents the geometrical term and the fiber coupling term, respectively. Recall the uniformity of optical intensity within the area of the receiving collimator, the total link loss in dB can be estimated by:

$$\eta = 10log_{10}(\alpha\beta) = 10log_{10}\left(\frac{(\varphi_c/2)^2}{R_{beam}^2}\frac{R_{core}^2}{\omega_f^2}\right), \tag{11}$$

where $R_{beam}$ is the beam radius at the receiving plane, and $R_{core}$ is the core radius of the coupling optical fiber, substituting Equation (8) into Equation (11), we have:

$$\eta = 20log_{10}\left(\frac{R_{core}}{f\theta}\right), \tag{12}$$

where the unit of $\theta$ is rad. With the objective of achieving a larger divergent angle $\theta$, the total link loss can be alleviated by either reducing the focal length of the receiving collimator or increasing the radius of the fiber core coupled to the receiving collimator. Nevertheless, a smaller focal length generally implies a collimator with a reduced CA, thereby leading to an increased geometrical path loss. Instead of optimizing Equation (12), a more effective approach to mitigating the total link loss involves compensating for the focus shift and the selection of a receiving collimator featuring a larger CA. Then, the total link loss becomes:

$$\eta = 10log_{10}\alpha = 20log_{10}\left(\frac{\varphi_c}{2z\theta}\right) \tag{13}$$

In addition, the core radius of the coupling fiber should be larger than the practical beam waist of the receiving collimator. As a result, the fiber coupling loss is eliminated, and the achievable ROP is much increased.

## 4. Experimental Results

### 4.1. Validity of the Gaussian Propagation Model

Figure 6 shows the experimental setup for measuring the ROPs of the divergent Gaussian beam. In the experiment, the half-divergent angle of the Gaussian beam is $10°$ and the transmission distance is 2.5 m. At the receiver, a zoomable receiving collimator is fixed on a laterally fixed rail, and the ROPs of different receiver angles are obtained by

translating the receiving collimator along the rail axis. Figure 7 showcases a comparison between the numerically simulated transverse ROP distribution and the experimentally measured discrete ROP values at intervals of 1°. The theoretically predicted curve and the experimentally measured curve with linear interpolation exhibit the same trend, with the ROP variance between them not exceeding 1 dB at any receiving angle. The result verifies the validity of employing geometric optics and uniform reception to describe the propagation of Gaussian beams in terms of intensity within short-range indoor OWC. Since the simulation only considers geometrical path loss, it can be inferred that the fiber coupling loss is effectively eliminated through the compensation of focus shifts. It is important to note that due to the narrow FOV of the receiving structure, the orientation of the collimator needs to be adjusted accordingly to align transceivers at varying transverse distances, potentially introducing ROP errors. Furthermore, due to constraints in the length of the rail, only positive transverse distances along the lateral axis were measured. Nevertheless, the spatial intensity distribution of Gaussian beams has a circular symmetry. However, the inhomogeneity of the ROP is as high as 10 dB.

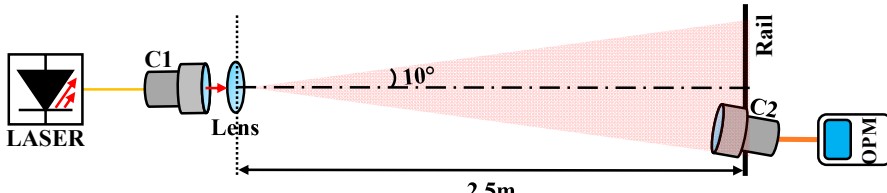

**Figure 6.** Experimental setup for measuring ROP distribution of a divergent Gaussian beam. C1: Emitting collimator; C2: Receiving collimator; OPM: Optical power meter.

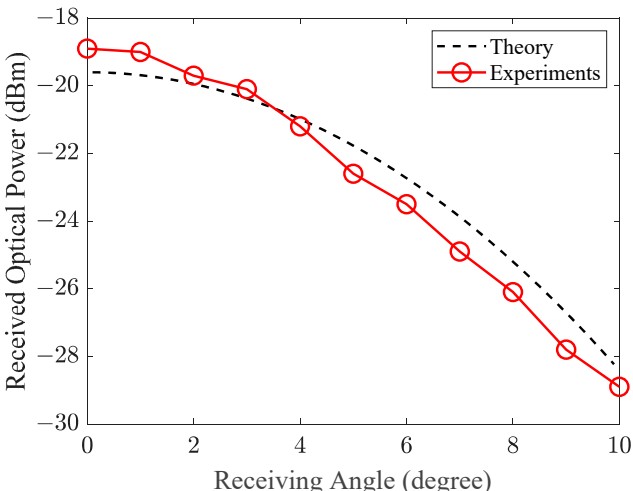

**Figure 7.** Theoretical and experimental ROP distribution of Gaussian beams.

### 4.2. MMF-Based Optical Beam-Shaping

Altering the spatial intensity distribution of a Gaussian beam offers an intuitive approach to addressing inhomogeneities. In this context, a segment of MMF is introduced between the SMF pigtailed laser source and the emitting collimator. By intentionally perturbing the transmission matrix of the MMF, the optical beam can be adaptively focused on any desired position within the coverage area. To assess the bit error rate (BER) performance across the coverage area, an experimental setup, as depicted in Figure 8, is implemented utilizing the MMF-based optical transmitter.

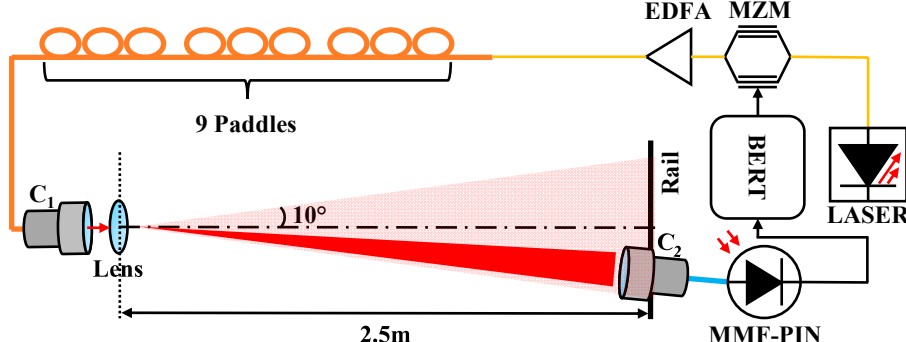

**Figure 8.** The MMF-based optical transmitter in OWC. BERT: Bit-error tester; MZM: Mach–Zehnder modulator; EDFA: Erbium-doped fiber amplifier; C1: Emitting collimator; C2: Receiving collimator; MMF-PIN: multimode-fiber PIN.

Initially, a Mach–Zehnder modulator (MZM, Fujitsu, FTM7937EZ) modulates the 10 Gbps baseband signal generated by a bit-error tester (BERT, 10 Gbps multi-channel BER tester, Luceo) onto the 1550 nm optical beam emitted by the laser source (Santec, MLS-2100). The modulated optical signal is then amplified through an Erbium-Doped Fiber Amplifier (EDFA, Amonics, AEDFA-BO-13) to 10 dBm. Next, the signal is directed into a 5 m long multi-mode patch cable (OM1 62.5/125 μm graded-index, Thorlabs, GIF625) with a radial offset distance of around 30 μm. The MMF is carefully wound around three three-paddle polarization controllers (nine control units in total, Thorlabs, FPC560). The optimized ROP is obtained by sequentially adjusting the rotation angle of each paddle in iterations. Then, the modulated optical signal is emitted into free space via the emitting collimator (Thorlabs, F810FC-1550) and the divergent lens (Thorlabs, AL2520M-C). After 2.5 m free-space transmission, the optical beam covers an area of 0.61 m$^2$, and a receiving collimator (Thorlabs, C40FC-C) is applied to capture the incoming light and then project it into another multi-mode patch cable (OM4 50/125 um graded-index, Thorlabs, GIF50E). The light coupled by the OM4 multi-mode patch cable is directly input into an MMF-PIN device (Thorlabs, DXM12DF), enabling optical-to-electrical (O/E) conversion. The received electrical signal is then compared to the transmitted electrical signal using the BERT, allowing for the counting of error bits. The optimization process for MMFs is iterative, requiring the measured ROP at each iteration to be transmitted back to the transmitter side until the desired target ROP is achieved. The optimized ROPs from 0 to 10° (1° interval) at a transmission distance of 2.5 m are sequentially measured.

Figure 9a shows the great homogenizing ability of MMF-based optical transmitters, and the optimized ROP variance within the coverage area is less than 2 dB. In contrast, SMF-based transmitters exhibit a significant ROP variance of 10 dB, owing to the Gaussian-shaped intensity distribution. Furthermore, at a 0 degree receiving angle, a notable ROP gain of 5.9 dB is observed, with the gain further increasing as the receiving angle expands. At a receiving angle of 10°, the ROP gain reaches 12.9 dB. This enhanced ROP substantially reduces the sensitivity requirements of direct detection receivers. Using a simple PIN-TIA photodiode as the O/E device without any optical pre-amplification, the BER performance of a 10 Gbps LOS OWC system through 2.5 m free-space transmission (Receiving angle = 8°) and an optical back-to-back link without free-space transmission are compared in Figure 9b. Remarkably, the MMF-based beam-shaping approach has a negligible ROP penalty (<0.2 dB), and the penalty may mainly caused by optical distortion at the edge of receiving lenses. We can conclude that the proposed scheme enables a 10 Gbps error-free transmission over an angular coverage of 10° × 10° at 2.5 m.

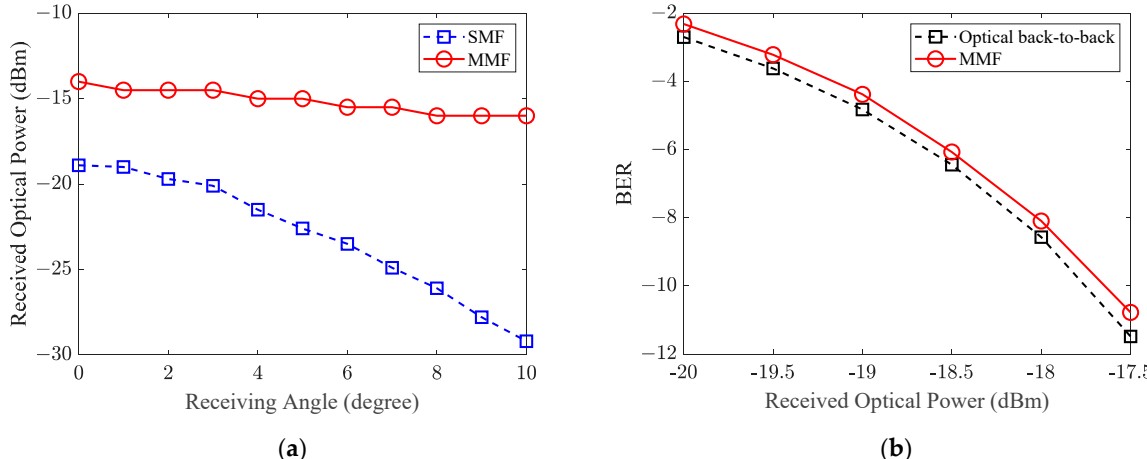

**Figure 9.** Performance comparison. (**a**) Comparison of ROP distribution between SMF and MMF; (**b**) Comparison of BER performance between optical back-to-back and MMF.

## 5. Conclusions

In the context of LOS OWC employing Gaussian divergent beams and narrow FOV receivers, a comparison is presented between the theoretical and experimental distributions of ROPs. The small discrepancy observed between the simulation and experimental results verifies the validity of geometric optics and uniform reception principles. Guided by these fundamental principles, the analysis focuses on the two major link losses inherent in the proposed system. Experimental results demonstrate the successful elimination of fiber coupling loss and the effective mitigation of geometrical path loss. Furthermore, a homogeneous ROP distribution, deviating from the Gaussian profile, is attained by manipulating MMFs, facilitating the mobility of end-users in indoor OWC. More importantly, the proposed scheme leads to a reduction in the complexity of tracking and steering operations.

## 6. Discussions

In future endeavors, the further optimization of link losses can be pursued by minimizing the divergent angle of the optical beam after the localization process. Leveraging classical iterative optimization algorithms and sufficient number of control units, multi-user access can be achieved utilizing a single fixed laser source. In addition, the enhanced mobility of portable devices can be achieved utilizing receivers with a larger FOV. However, it is important to acknowledge that the manipulation of MMFs requires iterations based on feedback loops, resulting in time-consuming optimization. To shorten the optimization time, the manipulation of MMFs should exploit their inherent properties to expedite the process. Lastly, the perturbations of MMFs are introduced by mechanical forces in the proposed scheme, and the inertia of these moving parts will limit the achievable rate of MMF optimizations.

**Author Contributions:** Conceptualization, Z.C.; methodology, X.Y.; software, Y.W.; validation, X.Y., Y.W.; data curation, X.Y.; writing—original draft preparation, X.Y.; writing—review and editing, E.T., Z.C., F.L. and C.L.; visualization, X.Y.; supervision, E.T.; project administration, Z.C., F.L. and C.L.; funding acquisition, E.T. All authors have read and agreed to the published version of the manuscript.

**Funding:** This research has been financed by Dutch Research Council NWO Gravitation Nanophotonics (Grant No. 024.002.033).

**Institutional Review Board Statement:** Not applicable.

**Informed Consent Statement:** Not applicable.

**Data Availability Statement:** Not applicable.

**Conflicts of Interest:** The authors declare no conflict of interest.

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
