# Peer review of "Two-Stage Link Loss Optimization of Divergent Gaussian Beams for Narrow Field-of-View Receivers in Line-of-Sight Indoor Downlink Optical Wireless Communication (Invited)"

_photonics, doi:10.3390/photonics10070815_

Round 1

Reviewer 1 Report

This work present a comparision tbetween theorical and experimental distributions of ROP considering OWC employing Gaussian divergent beams and narrow FOV receivers.

I think this work can be suitable for readers of this journal after  revisions as follows:

1.   Please clear the novelty of this work.

2.  Show equations (1-13) in fractional format: frac{}{}

3. Figure 2 could be represented in only one figure.

4. Write a paragraph that shows an introduction to the sections of the article

5. Describe the name of the manufacturer of the materials used in the experiment.

6. In Equations 11-13 write the base 10 logarithm using subscripts log_{10}

7. What disadvantages are there when implementing this proposal?

8. In line 120 add:  Equation (4).

9. Rewrite the equation (7).

10. Describe the Highlights of this work in the abstract.

11. Specify that the experiment was done in LOS

 It is seemed that some sentences need grammatically revision.

Author Response

We are honored to receive valuable feedback from the editor and reviewers, and the manuscript is carefully modified according to the comments one after another.

The main modifications are organized as follows:

  • The content of this article is expanded to 4000 words.
  • Sections of contents and discussions are added.
  • The highlights of this work is rearranged in the abstract.
  • Equations are represented in a formal format.
  • The manufacturers and part numbers of the materials used in experiments are described in details.

The Novelty of the work:

  • Divergent optical beam with modified spatial distributions.
  • Comprehensive link loss analysis and optimization.
  • Wavefront shaping using Multimode fibers (MMFs).
  • Homogenized ROP distribution by excitation of HOMs.

Disadvantages:

  • Limited response time due to mechanicals.
  • Limited optimization time due to iterative searching.

Reviewer 2 Report

The authors present a theoretical and experimental  study in  a non-return-to-zero on  off-keying modulation in intensity modulation and direct detection, where experimental results demon  strate a 10 Gbps error-free OWC downlink transmission at 2.5-m   without using any optical pre-amplifications at receivers.  The use  of visible and infrared (IR) light spectrum offers a vast range of over 300 THz of  unlicensed bandwidth, presenting an abundant resource.

This is an important topic of reserarch  today.  In the context of indoor OWC, whether operating in the visible or IR band, cost-effectiveness is a crucial consideration, emphasizing the need to avoid  complex devices and high computational requirements

The present study is well conducted  and discussed, and certainly will have impact in the área.  It is ready for publication.

xxx

Author Response

We are honored to receive valuable feedback from the editor and reviewers, and the manuscript is carefully modified according to the comments one after another.

The main modifications are organized as follows:

  • The content of this article is expanded to 4000 words.
  • Sections of contents and discussions are added.
  • The highlights of this work is rearranged in the abstract.
  • Equations are represented in a formal format.

The manufacturers and part numbers of the materials used in experiments are described in details.

Reviewer 3 Report

The manuscript addresses an important topic of
two-stage link loss optimization for divergent Gaussian beams. The presentation is scientifically sound and beautifully written. It adds value to the scientific community and is easy to read. The conclusions are clear and concise, and recommendations for future work are also presented. I recommend the paper to be published in its present form.

Author Response

(The authors gave the same response as above.)
